# Accelerated Osteogenic Differentiation of MC3T3-E1 Cells by Lactoferrin-Conjugated Nanodiamonds through Enhanced Anti-Oxidant and Anti-Inflammatory Effects

**DOI:** 10.3390/nano10010050

**Published:** 2019-12-24

**Authors:** Sung Eun Kim, Somang Choi, Jae-Young Hong, Kyu-Sik Shim, Tae-Hoon Kim, Kyeongsoon Park, Suk-Ha Lee

**Affiliations:** 1Department of Orthopedic Surgery and Rare Diseases Institute, Korea University Guro Hospital, #148, Gurodong-ro, Guro-gu, Seoul 08308, Korea; sekim10@korea.ac.kr (S.E.K.); chlthakd1029@naver.com (S.C.); 2Department of Biomedical Science, College of Medicine, Korea University, Anam-dong, Seongbuk-gu, Seoul 02841, Korea; breakdown88@korea.ac.kr; 3Department of Orthopedic Surgery, Korea University Ansan Hospital, #123, Jeokgeum-ro, Danwon-gu, Ansan-si, Gyeonggi-do 15355, Korea; osspine@korea.ac.kr; 4Department of Orthopedic Surgery, Konkuk University Medical Center, Konkuk University School of Medicine, #120-1, Neungdong-ro, Hwayang-dong, Gwangjin-gu, Seoul 05030, Korea; 5Department of Systems Biotechnology, College of Biotechnology and Natural Resources, Chung-Ang University, 4726 Seodong-daero, Daedeok-myeon, Anseong-si, Gyeonggi-do 17546, Korea; kspark1223@cau.ac.kr

**Keywords:** nanodiamonds, lactoferrin, anti-oxidant, anti-inflammatory, osteogenic differentiation

## Abstract

The purpose of this study was to investigate the effects of lactoferrin (LF)-conjugated nanodiamonds (NDs) in vitro on both anti-oxidant and anti-inflammation activity as well as osteogenic promotion. The application of LF-NDs resulted in sustained release of LF for up to 7 days. In vitro anti-oxidant analyses performed using Dichlorofluorescin diacetate (DCF-DA) assay and cell proliferation studies showed that LF (50 μg)-NDs effectively scavenged the reactive oxygen species (ROS) in MC3T3-E1 cells (osteoblast-like cells) after H_2_O_2_ treatment and increased proliferation of cells after H_2_O_2_ treatment. Treatment of lipopolysaccharide (LPS)-induced MC3T3-E1 cells with LF-NDs suppressed levels of pro-inflammatory cytokines, including interleukin-1β (IL-1β) and tumor necrosis factor-α (TNF-α). In addition, LF-NDs were associated with outstanding enhancement of osteogenic activity of MC3T3-E1 cells due to increased alkaline phosphatase (ALP) and calcium deposition. Our findings suggest that LF-NDs are an important substrate for alleviating ROS effects and inflammation, as well as promoting osteogenic differentiation of cells.

## 1. Introduction

Bone, plays major roles in the support, movement and protection of bodily organs and is a dynamic tissue with renewal and repair. Despite these properties, bone defects that result from trauma, traffic accidents (TA), congenital deformation, disease, and fracture sometimes require bone grafts. Autografts are considered the gold standard of bone graft replacements, due to their osteoinductive, osteoconductive and osteogenic properties. However, they also have shortcomings such as restricted supply, donor-site morbidity and accompanying pain [1,2]. Other options for treating bone defects are allografts, which are characterized by osteoinductive and osteoconductive characteristics. Major drawbacks of allografts are associated with blood loss, disease transmission, and cost [3,4].

The activities of osteoblasts and osteoclasts control bone remodeling via bone formation and resorption. However, there have been reports that inhibition of osteoblast differentiation and induction of apoptosis may occur due to oxidative stressors such as reactive oxygen species (ROS) [5,6,7]. In addition, the elevated levels of ROS in osteoblasts are associated with inhibition of mineralization and delayed bone healing [8,9,10].

Lactoferrin (LF), an iron-binding glycoprotein that is isolated from human and bovine milks, belongs to the transferrin family [11]. Oral administration of LF influences iron absorption and metabolism [12]. LF production increases in an inflammatory circumstance, which in turn indicates that LF has anti-inflammatory activity. LF is known as an antioxidant protein that increases antioxidant capacity and reduces ROS formation [13,14,15]. LF has pleiotropic effects including immunomodulatory, anticancer, antibacterial and antiviral activities [16,17,18]. Interestingly, LF is known to have potential as an osteogenic factor. Previous reports have demonstrated that materials functionalized with LF induce osteogenic differentiation of mesenchymal stem cells (MSCs) and adipose-derived stem cells (ADSCs) [19,20]. The mechanism of LF action on osteogenic differentiation stimulates cells via lipoprotein receptor-related protein 1 (LRP1)-independent protein kinase A (PKA) and p38 signaling pathways [21]. Therefore, to improve its therapeutic efficacy, LF should be incorporated or immobilized within/on substrates.

During the last 10 years, nanomaterials have been utilized in biomedical applications [22,23]. Among these, carbon-based nanomaterials such as carbon nanotubes, fullerenes, graphene and nanodiamond are major components of all living things [24]. Nanodiamond (ND) is considered an excellent material due to its good biocompatibility, low toxicity, hardness, and high surface functionality [24,25]. Moreover, ND can be facilitated to anchor drugs or biomolecules in different environments for delivery and sustained release due to non-covalent or covalent interactions [26,27]. Multifunctional ND can be used as a bone scaffold in combination with different polymers. Bone scaffolds fabricated by Zhang et al. [27] using PLLA polymer and octadecylamine-functionalized NDs (ODA-NDs) showed no harmful results on cell proliferation when murine osteoblasts were cultured on bone scaffolds for more than 1 week. Following this investigation, such scaffolds led to enhancement of biomechanical properties, such as a 280% increase in failure strain and a 310% improvement of the fracture force in tensile strength [28]. In other studies performed by Parizek et al. [29] and Ahn et al. [30], ND-composited PLGA nanofibrous membranes and PCL fibrous matrices not only enhanced mechanical resistance but also induced proliferation and differentiation of osteoblast-like cells.

In the present study, we designed NDs-based nanoparticles with anti-oxidant, anti-inflammatory and osteogenic effects by anchoring LF. Here, we demonstrate the utility of multi-functional ND with LF by anti-oxidant, IL-1β and TNF-α assays, alkaline phosphatase (ALP) activity, and calcium content against MC3T3-E1 cells.

## 2. Materials and Methods

### 2.1. Lactoferrin (LF) Conjugated Carboxylated-Nanodiamonds (cNDs)

For the conjugation of human lactoferrin (LF, Sigma-Aldrich, St. Louis, MO, USA), 10 mg of carboxylated-nanodiamonds (cNDs, Tokyo Chemical Industry Co., Ltd., Tokyo, Japan) were placed in a sterile PBS solution (pH 7.4) and gently stirred for 30 min at room temperature (RT), followed by the addition of LF (10 or 50 μg·mL^−1^) and incubated for 24 h. After incubation for 24 h, the product was rinsed three times with PBS solution and centrifuged at 3000 rpm for 10 min at 4 °C using Micro Refrigerated Centrifuge (Smart R17, Hanil Science Industrial, Incheon, Korea). The supernatant was collected to analyze the loading amount of LF on cND surface and the sample was lyophilized using freeze dryer (FD8508, IlShinBioBase Co., Ltd. Gyeonggido, Korea) for 3 days. To quantify the loading amount of LF, the collected supernatant after the LF conjugated cNDs was measured with a Pierce bicinchoninic acid (BCA) protein assay kit (Thermo Fisher Scientific, Rockford, IL, USA) following the manufacturer’s protocol. In brief, the supernatant (25 μL) was placed in e-tube containing 200 μL of BCA reagent and incubated at RT for 1 h at 37 °C. After incubation, 100 μL of solution was transferred to 96-well plates and monitored at 562 nm using a Flash Multimode Reader (Varioskan^TM^, Thermo Scientific, Waltham, MA, USA). The loading amount of LF on the ND surface was evaluated by determining the loading amount of LF relative to the initial LF amount. cNDs, LF (10 μg) conjugated cNDs, and LF (50 μg) conjugated cNDs were designated as NDs, LF (10 μg)-NDs and LF (50 μg)-NDs, respectively.

### 2.2. Characterization

Morphologies of NDs with or without conjugated LF were visualized with a transmission electron microscope (TEM, JEM-F200, JEOL Ltd., Tokyo, Japan) at the Yonsei Center for Research Facilities. Prior to the observation of TEM, each sample was pre-treated as follows: 10 μg of each sample was dispersed in an e-tube containing 1 mL ethanol (EtOH), followed by sonication using Powersonic 405 (bath-type instruments; 40 KHz, power: 350 W, Hwashin Tech Co., Ltd., Seoul, Korea) for 1 h at 4 °C. Each sample was then pipetted onto a copper TEM grid (CF200-Cu, Electron Microscopy Sciences, Hatfield, PA, USA) and the solvent of each sample was removed after overnight deposition. TEM was accelerated at 200 kV.

To confirm the size distribution of each sample, 100 μg·mL^−1^ of each sample was suspended using deionized and distilled water (DDW) and sonicated using a Powersonic 405 for 1 h at 4 °C. Then, 1 mL of each dispersed sample was pipetted onto cuvettes (CUVEETTA STD UV 4 FACCE OTT. Kartell S.p.A., Noviglio, Italy). Dynamic light scattering (DLS) analysis was conducted to assess the size distribution of each sample using a Malvern Matersizer 3000 instrument (Malvern Panalytical Ltd., Malvern, UK) with a He-Ne laser at a wavelength of 633 nm. For zeta potential measurements, 1 mL of each dispersed sample was placed in a disposable micro-cuvette (Zetasizer Nano Series, Malvern Panalytical Ltd.). The zeta potential measurements were also performed on Malvern Matersizer 3000 instrument. In order to observe the surface chemical compositions of NDs with or without LF, we performed X-ray photoelectron spectroscopy (XPS) with a K-alpha spectrometer (ESCALAB250 XPS System, Theta Probe AR-XPS System, Thermo Fisher Scientific, Waltham, MA, USA) with 1486.6 eV photons as an Al Kα X-ray source at the Korea Basic Science Institute Busan Center. The surface compositions of different test groups were analyzed using attenuated total reflectance Fourier transform infrared (ATR-FTIR, Avatar 360, Nicolet Instrument Corp., Madison, WI, USA) spectroscopy with a resolution of 4 cm^−1^ between 4000 and 600 cm^−1^. The ATR-FTIR spectrum of LF alone was observed to measure a commercially available powder.

### 2.3. In Vitro LF Release

In order to assess the release of LF from LF (10 μg)-NDs and LF (50 μg)-NDs, we performed assays at pre-designated time intervals. At each interval, 10 mg of each sample was dispersed directly in 1 mL of PBS solution (pH 7.4) and pipetted in a dialysis bag (cutoff molecular weight 6000–8000 Da). The bag was placed in a conical tube containing 5 mL PBS, followed by shaking at a speed of 100 rpm at 37 °C. PBS solution used as a release medium was harvested at pre-designated time intervals and replaced by the same volume of fresh PBS. The amount of LF released was monitored with a Pierce BCA protein assay kit using a Flash Multimode Reader at 562 nm.

### 2.4. Cytotoxicity

Before the determination of the effect of LF-NDs on the cellular activity, we performed cytotoxicity tests of all samples. Briefly, MC3T3-E1 cells (Korean Cell Line Bank, Seoul, Korea) were seeded at a density of 1 × 10^4^ cells per well in 96-well plates and cultured with Dulbecco’s modified Eagle’s medium (DMEM, Thermo Fisher Scientific Inc., USA) containing 10% fetal bovine serum (FBS, Thermo Fisher Scientific Inc., USA) and 1% antibiotics (100 U·mL^−1^ penicillin and 0.1 mg·mL^−1^ streptomycin, Thermo Fisher Scientific Inc., USA) in 5% CO_2_ at 37 °C. After culturing for 24 h, cells were washed with PBS and exposed to each sample (100 μg·mL^−1^). After cultivation for 24 and 48 h, DMEM in cells was aspirated and PBS buffer was added to wash cells. Then, 10 μL of 1-(4,5-dimethylthiazol-2-yl)-3,5-diphenyl formazan (MTT) reagent (Sigma-Aldrich) was added to each well and incubated for 4 h in the dark at 37 °C. At the end of the incubation period, 150 μL of dimethyl sulfoxide (DMSO, Sigma-Aldrich, St. Louis, MO, USA) was added to each well to dissolve the formazan crystals and absorbance was read at 595 nm with a Flash Multimode Reader. Medium from cells without samples were used. The cytotoxicity was represented as the percentage of viable cells vs. the control.

### 2.5. Cellular Uptake Anaalysis

In order to assess the cellular uptake of cNDs, cNDs were conjugated with fluorescein isothiocyanate (FITC, Thermo Fisher Scientific, USA). Before FITC conjugation, the cND surface was first modified by dopamine (Sigma-Aldrich) to anchor the amine group. Briefly, cNDs (10 mg) were suspended in 10 mL of 10 mM Tris·HCl (pH 8.0), dissolving 10 mg of dopamine, and the mixture was gently stirred while avoiding light exposure for 24 h. After reaction, NH_2_-NDs were washed several times with DDW and lyophilized for 3 days. Ten mg of NH_2_-NDs were dispersed in MES buffer (pH 4.5) and 100 μg of FITC was added to the buffer. Mixtures were maintained at RT in the dark overnight. The products were centrifuged, and the sediments were rinsed with DDW and lyophilized for 2 days. Cells at a density of 1 × 10^4^ cells per dish were seeded on microscope cover glasses (12 mm diameter, Paul Marienfeld GebH & Co., Lauda-Königshofen, Germany) and adhered for 24 h. After adhesion for 24 h, cells were rinsed twice with PBS and exposed to FITC-conjugated NDs (100 μg·mL^−1^), followed by incubation at 37 °C and 5% CO_2_ for 4 h. After incubation, cells were fixed with 4% paraformaldehyde for 30 min. Rhodamine-phalloidin (1:200, Thermo Fisher Scientific, USA) and 4-6-diamidino-2-phenylindole (DAPI, Thermo Fisher Scientific, USA) were added to cells for staining cell nuclei for 30 min at RT. Then, the samples were observed by a confocal laser scanning microscope (CLSM, LSM700, Zeiss, Germany).

### 2.6. Evaluation of ROS Scavenging Activity

#### 2.6.1. Suppression of ROS at the Cell Level

In order to measure the ROS scavenging capacity of LF-NDs in cells, we performed 2′,7-dichlorodihydrofluorescein diacetate (DCFDA) staining and DCFDA assays. MC3T3-E1 was seeded at a concentration of 1 × 10^4^ cells per well at microscope cover glasses in 24-well plates and incubated for 24 h. After incubation for 24 h, cells were treated with 300 μM H_2_O_2_ at 37 °C for 30 min, followed by the aspiration of 300 μM H_2_O_2_. Cells were treated using the extracted DMEM without FBS from each sample for 24 h. At pre-designed times of 6 or 24 h, the cells were stained with DCFDA (25 μM) for 45 min in the dark, rinsed with PBS, and fixed with 3.7% paraformaldehyde for 20 min. Cell images were observed using a confocal laser scanning microscope. In order to further quantify the ROS levels in cells served with the extracted DMEM from each group, cells were examined with a DCFDA/H2DCFDA cellular ROS assay kit (Abcam, Cambridge, MA, USA) in accordance with the manufacturer’s protocols. The quantitative fluorescence spectra were recorded by a Flash Multimode Reader with excitation/emission at 495 nm/529 nm.

#### 2.6.2. Protection of Cell Suppression in the ROS Condition

Cell survival capacity of MC3T3-E1 cells treated with each sample in ROS condition was analyzed using MTT reagent. The cells were seeded in 24-well plates at a density of 1 × 10^5^ cells per well and incubated with DMEM in the presence or absence of each sample (100 μg·mL^−1^) for 24 h. After 24 h, the cells were exposed to 300 μM H_2_O_2_ at 37 °C for 30 min, followed by another incubation for 6 or 24 h. The cells were treated with MTT reagent for 4 h at 37 °C in 5% CO_2_. After 4 h incubation, formazan crystals were formed, followed by the addition of DMSO to dissolve the formazan crystals. The solution was added to 96-well plates and monitored at 595 nm with a Flash Multimode Reader.

### 2.7. Interleukin-1β (IL-1β) and Tumor Necrosis Factor Alpha (TNF-α) Content

In order to assess the anti-inflammatory activities of LF-NDs, 1 × 10^5^ cells were seeded in each well of a 24-well plate with DMEM and treated with 100 μg·mL^−1^ of LPS in the presence or absence of each sample (100 μg·mL^−1^). At pre-designed time points, the supernatants were harvested and stored at −20 °C for further quantification of IL-1β and TNF-α. The amount of IL-1β and TNF-α secreted in cells was analyzed using enzyme-linked immunosorbent assay (ELISA) kits (BioGems Ltd., Westlake Village, CA, USA). The absorbance value was monitored at 450 nm using a Flash Multimode Reader.

### 2.8. Alkaline Phosphatase (ALP) Activity

MC3T3-E1 cells (1 × 10^5^ cells·mL^−1^) were seeded on 24-well plates and exposed to each sample at a concentration of 100 μg·mL^−1^. At the end of each period, cells were lysed with lysis buffer (1× RIPA buffer) and transferred to e-tubes. Cell lysates were centrifuged at 13,500 rpm for 10 min at 4 °C using a Micro Refrigerated Centrifuge (Smart R17, Hanil Science Industrial, Incheon, Korea). Supernatant was transferred to new e-tubes and P-nitrophenyl phosphate (Sigma-Aldrich, USA) solution was added, followed by incubation at 37 °C for 30 min. After incubation for 30 min, 500 μL of 1N NaOH was added to the solution to stop the reaction. The absorbance was evaluated at 405 nm with a Flash Multimode Reader. Total protein concentration was normalized using Bradford reagent (Bio-Rad Laboratories, Inc., Hercules, CA, USA) and bovine serum albumin (BSA, Bio-Rad Laboratories, Inc.).

### 2.9. Calcium Deposition

MC3T3-E1 cells (1 × 10^5^ cells·mL^−1^) were seeded in 24-well plates and cultured with each sample (100 μg·mL^−1^). After being exposed to each sample at pre-determined time intervals, cells were rinsed three times with PBS and 500 μL of 0.5N·HCl was added to cells, followed by incubation at 100 rpm overnight at 37 °C using shaking incubator (SI-300R, Jeio Tech Co., Ltd., Seoul, Korea). After overnight incubation, each sample (20 μL) was transferred to an e-tube and calcium standard solution (20 μL) was added to the sample solution, following the addition of color reagent solution (400 μL) [25 mg of o-cresolphthalein complexone (Sigma-Aldrich) and 250 mg of 8-hydroxy-quindine (Sigma-Aldrich)]. The resulting solution was vortexed for 1 min and then 20 μL of AMP buffer [37.8 mL of 2-amino-2-methyl-1-propanol (Sigma-Aldrich)] was added and reacted for 15 min at RT. After the reaction, the solution (200 μL) was carefully transferred to 96-well plates and absorbance was recorded with a Flash Multimode Reader at 575 nm.

### 2.10. Statistical Analysis

Data are presented as mean ± standard deviation. Statistical comparisons were performed via one-way analysis of variance (ANOVA) using Systat software (Chicago, IL, USA). Differences were considered statistically significant at * *p* < 0.05 and ** *p* < 0.01.

## 3. Results

### 3.1. Characterization of NDs with and without LF

The TEM images in Figure 1 distinctly demonstrate the morphologies of NDs with and without LF. Each ND group was round shape and nano-sized. The diameters and size distributions of NDs, LF (10 μg)-NDs, and LF (50 μg)-NDs were investigated by DLS.

As shown in Figure 2A, the particle sizes and distributions were 209.00 ± 103.10 nm with a polydispersity index (PDI) of 0.203 for NDs, 211.30 ± 95.22 nm with PDI of 0.189 for LF (10 μg)-NDs, and 216.50 ± 108.50 nm with PDI of 0.183 for LF (50 μg)-NDs, respectively. Smaller hydrodynamic diameter and narrower PDI of LF-NDs were observed, which suggests that LF-NDs have improved dispersibility due to protein conjugating. The zeta potential values of NDs, LF (10 μg)-NDs, and LF (50 μg)-NDs were −26.23 ± 0.80, −27.77 ± 1.30 and −28.50 ± 0.85 mV, respectively.

XPS was conducted to confirm the surface chemical compositions of NDs, LF (10 μg)-NDs and LF (50 μg)-NDs (Table 1). NDs conjugated by LF (10 or 50 μg) were confirmed by increases in N1s component from 1.83% to 3.09% and 1.83% to 4.74, respectively, indicating that LF is existed on the surfaces of the NDs. To further confirm the LF immobilization on the NDs, ATR-FTIR spectra of each group are shown in Figure 2B before and after LF (10 or 50 μg) conjugation. After conjugating with LF, we observed strong absorption new peak bands at 1635 and 1517 cm^−^^1^, which correspond to the C=O stretching vibration of amide I and N-H bending vibration of amide II, respectively, suggesting the successful conjugation of LF. The loading amount and efficiency of LF from LF (10 μg)-NDs and LF (50 μg)-NDs were 6.44 ± 0.37 μg (64.39 ± 3.66%) and 41.15 ± 1.94 μg (82.30 ± 3.88%), respectively.

### 3.2. In Vitro LF Release

As shown in Figure 3, the in vitro release profiles of LF from LF (10 μg)-NDs and LF (50 μg)-NDs showed sustained release patterns. At 1 day, the released amounts and percentages of LF were 3.60 ± 0.08 μg (55.95 ± 0.57%) for LF (10 μg)-NDs and 25.06 ± 0.42 μg (60.90 ± 1.03%) for (50 μg)-NDs. For the 7-day period, LF (10 μg)-NDs and LF (50 μg)-NDs released 4.54 ± 0.29 μg (70.56 ± 4.44%) and 29.22 ± 0.41 μg (71.00 ± 0.99%) of LF, respectively.

### 3.3. Cytotoxicity and Cellular Internalization

Figure 4A shows the cytotoxicity test results for each sample compared against MC3T3-E1 cells at 24 and 48 h. Viabilities of cells treated with each sample were preserved over 98% for 48 h compared to the control group, suggesting that there were no cytotoxic effects on MC3T3-E1 cells in any sample. CLSM was used to confirm the intracellular uptake of NDs with or without LF. Previous study showed that ND particles can be internalized through the cell membrane and accumulate in the cytoplasm [31]. Consistent with the previous results, after 4 h incubation, FITC-conjugated NDs were observed around the cytoplasm and nuclei of cells (Figure 4B).

### 3.4. ROS Scavenging Effects of LF-NDs in Cells

In order to investigate the anti-oxidant activities of each sample, MC3T3-E1 cells were pre-treated with 300 µM H_2_O_2_ exposure for 30 min in order to create ROS. Under 300 µM H_2_O_2_ condition, controls without sample treatment showed high fluorescence intensity in images taken at 6 and 24 h (Figure 5A,B). However, cells treated with extracts of NDs with or without LF showed low fluorescence intensities and images in a time-dependent manner. Treatment with extract from LF (50 μg)-NDs led to the lowest fluorescence intensity and images among groups. These results indicate that LF (50 μg)-NDs have excellent anti-oxidant activity.

### 3.5. Cellular Protection Against ROS

In order to further demonstrate the direct anti-oxidant effects of LF-NDs in cells, we measured the proliferation of MC3T3-E1 cells treated with 300 μM H_2_O_2_ in the presence or absence of each sample at 6 and 24 h. As shown in Figure 6, there were significant differences in cell proliferation between MC3T3-E1 cells treated with NDs with or without LF and those of controls at 6 and 24 h, while the cell viability of the control group was reduced to 24 h rather than 6 h. However, viabilities of cells treated with LF-NDs were greater than those treated with NDs in a concentration- and time-dependent manner.

### 3.6. Levels of Pro-Inflammatory Cytokines in Cell Supernatants of LPS-Induced MC3T3-E1 Cells Treated with LF-NDs

Figure 7 shows the levels of pro-inflammatory cytokines, including IL-1β and TNF-α, in cell supernatant secreted by LPS-induced MC3T3-E1 cells in the presence or absence of each sample at pre-designated time intervals of 2, 6, 24, 72 and 120 h. The levels of pro-inflammatory cytokines in MC3T3-E1 cells treated with LPS significant increased compared to untreated cells in a time-dependent manner. However, the treatment of MC3T3-E1 cells with LPS including NDs with or without LF reduced cytokines of IL-1β and TNF-α compared with those of MC3T3-E1 cells with LPS treatment. When comparing NDs with LF and NDs, we noted significant suppression of IL-1β and TNF-α. Moreover, the cytokines IL-1β and TNF-α decreased significantly in MC3T3-E1 cells treated with LF (50 μg)-NDs compared to LF (10 μg)-NDs.

### 3.7. Alkaline Phosphatase (ALP) Activity and Calcium Deposition

To assess whether NDs with or without LF are effective for the differentiation of MC3T3-E1 cells, we measured ALP activity at 3 and 7 days. As shown in Figure 8A, the in vitro ALP activities of MC3T3-E1 cells treated with all test samples increased gradually in a time-dependent manner. The addition of NDs with LF significantly promoted ALP activity compared with NDs at 3 and 7 days. As expected, MC3T3-E1 cells treated with LF (50 μg)-NDs exhibited higher ALP activity than did cells treated with NDs and LF (10 μg)-NDs at 3 and 7 days. Generally, calcium deposition was measured as a marker of osteogenic differentiation, and upregulation of calcium deposition is a major event that occurs during late time points of osteogenesis [20,32]. Figure 7B shows the in vitro amounts of calcium deposited by MC3T3-E1 cells treated with different samples for different culture times. The amount of calcium deposited increases as incubation time intervals increase up to 21 days in all experimental groups. Calcium deposition by MC3T3-E1 cells treated with NDs containing LF was markedly higher than for MC3T3-E1 cells treated with NDs at 7 and 21 days. Moreover, significant differences in the amount of calcium deposited by MC3T3-E1 cells treated with LF (50 μg)-NDs vs. LF (10 μg)-NDs were observed at 7 and 21 days. These results supported that LF-NDs promoted the osteoblastic differentiation of MC3T3-El cells.

## 4. Discussion

Bone tissue undergoes continuous bone remodeling throughout life, in which bone resorption and bone formation are regulated by the parallel activity of osteoblasts and osteoclasts [33,34]. The bone remodeling cycle, in which the structure of the bone is organized in regular units and the mass of the bone gains maximum resistance to mechanical forces acting on the bone, entails three stages: (1) the initiation of osteoclasts to form and reabsorb damaged bone; (2) the conversion of osteoclasts into osteoclast activity; and (3) formation when osteoclasts replace a portion of the reabsorbed bone [35]. Hormonal imbalances or aging can lead to osteoporosis through the disruption of bone resorption and balance, which eventually increases the risk of bony fracture.

The purpose of this study was to investigate whether osteogenic differentiation of MC3T3-E1 cells on anti-oxidant and anti-inflammatory LF-NDs could be improved by fabricating lactoferrin-conjugated NDs. LF-conjugated NDs were fabricated via electrostatic interactions between amine groups of LF and carboxyl groups of NDs. NDs, LF (10 μg)-NDs and LF (50 μg)-NDs were observed by TEM to be approximately 200 nm in size. The particle sizes of each sample measured by DLS were also confirmed to be about 200 nm. In addition, NDs after conjugating LF showed an increase N1s content and a decrease C1s content in comparison to those of bare NDs when measured by XPS. As previously reported, N1s contents increased on heparin-porous microspheres (Hep-PMs) and heparin-titanium (Hep-Ti) after immobilizing LF, compared with PMs or Ti [20,32]. These previous results were consistent with our results in the present study, and indicate that successful LF conjugation on carboxylated NDs may be achieved by electrostatic interactions.

ROS, such as superoxide anion (O_2_^−^), hydrogen peroxide (H_2_O_2_), and hydroxyl radical (HO•), are oxygen-containing molecules that play a detrimental role in age-related diseases because their levels increase with age or the onset of inflammation [36]. The imbalance between ROS production and antioxidant mechanisms leads to oxidative stress affecting the bone, which eventually accelerates the destruction of calcified tissue and bone resorption. Hydrogen peroxide (H_2_O_2_), which has strong oxidizing properties and is formed by many oxidizing enzymes, can cross the membrane and oxidize many compounds slowly, and thus is widely used to induce oxidative stress in vitro [5,37].

In order to confirm the radical scavenging activity of each samples, we conducted indirect and direct assessments in cells exposed to H_2_O_2_ such as DCFDA assays and cell viability assays, respectively. For determination of the scavenging activity in each sample using indirect methods, cells were pre-treated with 300 μM H_2_O_2_ to stimulate oxidative stress, followed by treating extracts from each sample. As observed in the fluorescence assay and CLSM images, NDs with or without LF significantly decrease the fluorescence signal and images in cells compared to cells without sample treatment at 6 and 24 h. Moreover, fluorescence signals and images in cells treated with the extracts from LF-NDs decreased compared to those from NDs in a dose- and time-dependent manner. To further estimate the scavenging activity of all test groups using direct methods, we measured cell viabilities treated with 300 μM H_2_O_2_ condition in the presence or absence of each test group. The cell viabilities were diminished in a time-dependent fashion due to oxidative damage of cellular components by H_2_O_2_ stimulation [38,39]. However, treatments with NDs and LF-NDs significantly increased cell proliferation. In addition, LF-conjugated ND groups showed much higher cell viabilities than the ND groups and extended cell proliferation in a dose- and time-dependent manner. These results indicate that LF molecules conjugated to NDs on ND surfaces can effectively counter intracellular ROS and interfere with cell suppression through oxidative damage, thus increasing cell viability and proliferation.

Oxidative stress can induce an inflammatory response through activation of redox-sensitive transcription NF-κB and is known to play an important role in inducing inflammatory responses [6,8,37]. In the early stages of bone repair, pro-inflammatory cytokines are released from the site of injury, and such cytokines can slow bone repair. As reported previously, pro-inflammatory cytokines inhibit osteogenic differentiation from MSCs and ADSCs [40,41]. Therefore, we investigated the in vitro anti-inflammatory activities of LF-NDs in inflamed cells. In order to mimic the inflammatory environment in vitro, cells were treated with LPS which is a major component of the outer membrane of Gram-negative bacteria, also known as lipoglycans and endotoxins. As previously studied, LPS-stimulated cells secreted increased amount of pro-inflammatory cytokines, such as TNF-α, IL-6 and IL-1β [42,43]. In order to determine the in vitro anti-inflammatory activities of LF-NDs, MC3T3-E1 cells were treated with LPS in the presence or absence of NDs with or without LF to induce an in vitro inflammatory environment, followed by the acquirement of supernatant secreted by cells at predetermined time points and measurements of the pro-inflammatory cytokines (IL-1β and TNF-α) using ELISA. Treatment with LPS upregulated IL-1β and TNF-α levels, whereas cells treated with NDs with or without LF showed decreased IL-1β and TNF-α levels in a time-dependent manner. As expected, cells treated with NDs conjugating different LF concentrations showed lower IL-1β and TNF-α levels than did cells treated with bare NDs, due to the presence of LF, which is known for inhibiting the secretion of pro-inflammatory cytokines. Prior studies reported that LF inhibited pro-inflammatory cytokines including IL-1, IL-6 and TNF-α in a monocytic cell line (THP-1) stimulated by LPS [44,45]. Rasheed et al. [46] demonstrated that LF inhibited prostaglandin E_2_ (PGE_2_) production and cyclocoxygenase-2 (COX-2) expression in IL-1β-induced human osteoarthritis via suppression of NF-κB activation. Consistent with these studies, we found that LF-NDs suppressed pro-inflammatory cytokines, such as IL-1β and TNF-α.

The ALP activities and calcium deposition of NDs conjugating LF-treated MC3T3-E1 cells were markedly higher than for those treated with bare NDs in a dose- and time-dependent manner, because LF molecules released from NDs affected osteogenic differentiation [20,32,47]. These findings suggest that LF molecules conjugated on NDs can induce osteogenic differentiation of MC3T3-E1 cells by sustained release of LF compared with bare NDs.

Consequently, this study demonstrated that NDs as a delivery carrier effectively ferry LF into cells. Through the effective delivery of LF with pleiotropic effects such as anti-inflammatory and anti-oxidant properties, LF-NDs could exert a synergistic effect on the osteogenic differentiation of MC3T3-El cells. Therefore, we expect that the anti-inflammatory and anti-oxidant LF-NDs will be applicable to treat bone tissue regeneration.

## 5. Conclusions

In this study, LF-conjugated NDs were developed to investigate their effects against oxidative stress, inflammatory response and osteogenic differentiation of cells. LF-conjugated NDs were first fabricated by electrostatic interactions between amine groups of LF and carboxyl groups of NDs. LF-NDs not only effectively scavenge ROS in cells, but also protect cells in ROS environments and can significantly suppress the levels of pro-inflammatory cytokines (IL-1β and TNF-α) secreted by LPS-stimulated cells. In addition, LF-NDs induce osteogenic differentiation of MC3T3-E1 cells by enhancing ALP activity and calcium deposition via release of LF. Thus, LF-NDs exhibit superior capacities for enhanced anti-oxidant and anti-inflammatory functions as well as induced improved osteogenic differentiation of cells. LF-NDs have great potential for application in bone regeneration and disease treatment.

## Figures and Tables

**Figure 1 nanomaterials-10-00050-f001:**
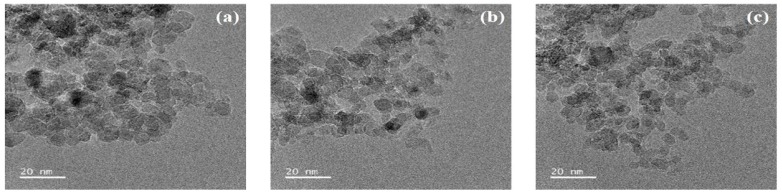
Transmission electron microscope (TEM) images of (**a**) NDs; (**b**) LF (10 μg)-NDs; (**c**) LF (50 μg)-NDs. Scale bar: 20 nm.

**Figure 2 nanomaterials-10-00050-f002:**
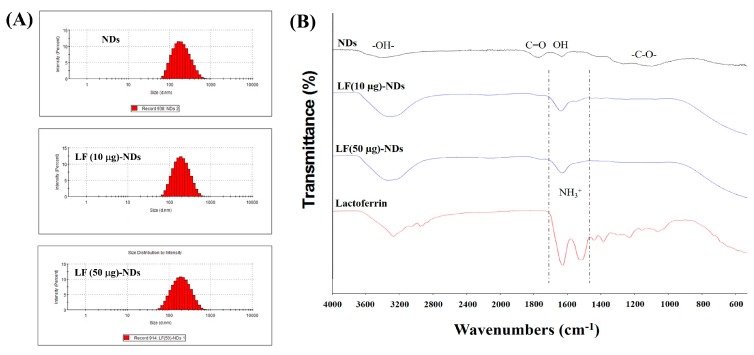
(**A**) Particle size and distribution of NDs, LF (10 μg)-NDs and LF (50 μg)-NDs measured by dynamic light scattering (DLS); (**B**) Attenuated total reflectance Fourier transform infrared (ATR-FTIR) spectra of NDs, LF (10 μg)-NDs, LF (50 μg)-NDs and Lactoferrin.

**Figure 3 nanomaterials-10-00050-f003:**
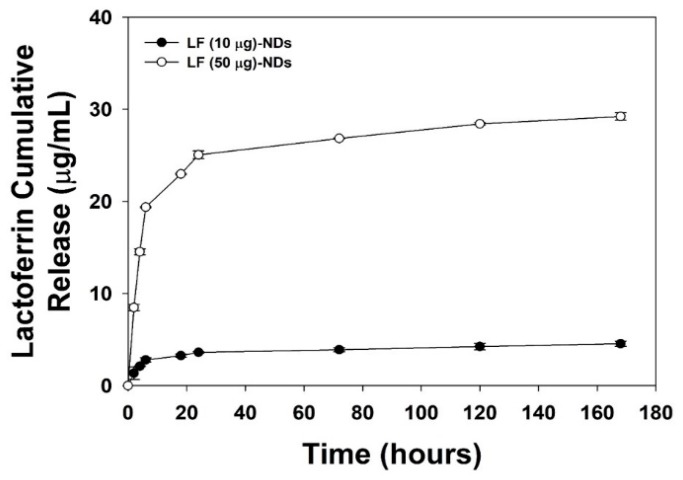
In vitro release profiles of LF from LF (10 μg)-NDs and LF (50 μg)-NDs for 7 days. (*n* = 4).

**Figure 4 nanomaterials-10-00050-f004:**
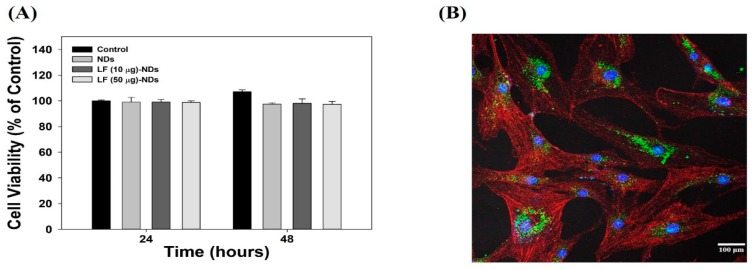
(**A**) Cytotoxicity test of NDs, LF (10 μg)-NDs and LF (50 μg)-NDs against MC3T3-E1 cells for 24 and 48 h; (**B**) In vitro cellular internalization of FITC-NDs after incubation of 4 h. Scale bar: 100 μm.

**Figure 5 nanomaterials-10-00050-f005:**
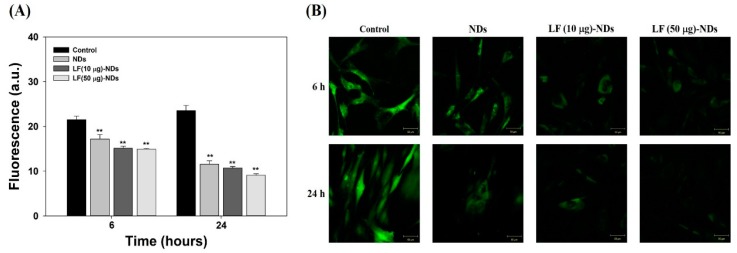
(**A**) Quantitative intracellular ROS levels of MC3T3-E1 cells treated with extract from NDs, LF (10 μg)-NDs and LF (50 μg)-NDs for 6 and 24 h after the cells were treated with 300 μM H_2_O_2_ for 30 min. The error bars represent mean ± standard deviation (*n* = 4). *p* value is a comparison between LF-conjugated NDs and NDs. ** *p* < 0.01; (**B**) Fluorescence images of intracellular levels of MC3T3-E1 cells treated with extract from NDs, LF (10 μg)-NDs and LF (50 μg)-NDs for 6 and 24 h after the cells were treated with 300 μM H_2_O_2_ for 30 min. After 6 and 24 h treatment, the cells were stained with 2′,7-dichlorodihydrofluorescein diacetate (DCFDA) and observed by a confocal laser scanning microscope (CLSM). Scale bar = 50 μm.

**Figure 6 nanomaterials-10-00050-f006:**
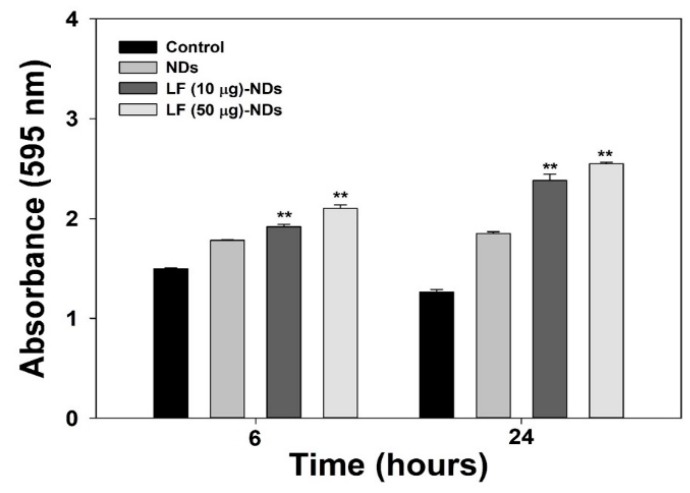
Cell viabilities of MC3T3-E1 cells treated with NDs with or without LF at 6 and 24 h after pre-treating under 300 μM H_2_O_2_ condition. The error bars represent mean ± standard deviation (*n* = 4). *p* value is a comparison between LF-conjugated NDs and NDs. ** *p* < 0.01.

**Figure 7 nanomaterials-10-00050-f007:**
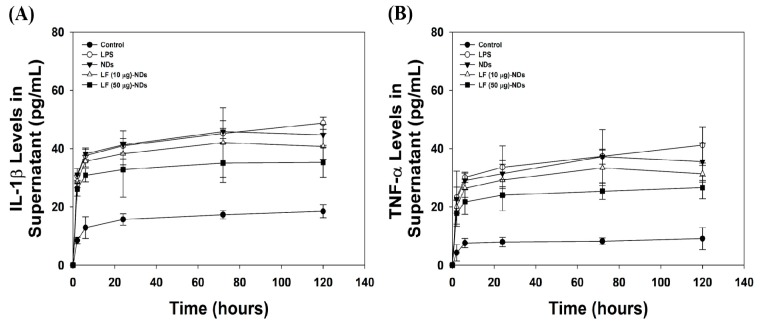
(**A**) IL-1β and (**B**) TNF-α levels in cell supernatant secreted by LPS-stimulated cells treated with NDs with or without LF at 2, 6, 24, 72 and 120 h by ELISA.

**Figure 8 nanomaterials-10-00050-f008:**
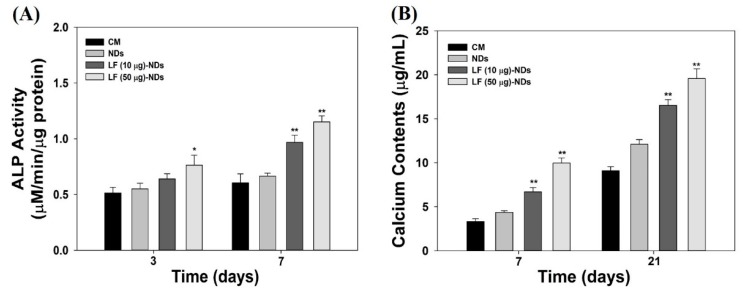
(**A**) Alkaline phosphatase (ALP) activity of MC3T3-E1 cells treated with LF (10 μg)-NDs and LF (50 μg)-NDs after 3 and 7 days of incubation; (**B**) Calcium deposition by ME3T3-E1 cells treated with NDs, LF (10 μg)-NDs and LF (50 μg)-NDs after 7 and 21 days of incubation. The error bars represent mean ± standard deviation (*n* = 4). *p* value is a comparison between LF-conjugated NDs and NDs. * *p* < 0.05 and ** *p* < 0.01.

**Table 1 nanomaterials-10-00050-t001:** Surface elemental composition of NDs, LF (10 μg)-NDs, and LF (50 μg)-NDs.

	Elements	C1s (%)	N1s (%)	O1s (%)	Total (%)
Sample					
NDs		87.32	1.83	10.85	100
LF (10 μg)-NDs		85.99	3.09	10.92	100
LF (10 μg)-NDs		84.45	4.74	10.81	100

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
