# Peer review of "Accelerated Osteogenic Differentiation of MC3T3-E1 Cells by Lactoferrin-Conjugated Nanodiamonds through Enhanced Anti-Oxidant and Anti-Inflammatory Effects"

_nanomaterials, 2019, doi:10.3390/nano10010050_

Round 1

Reviewer 1 Report

Nanodiamond (ND) particles with various terminations for  medical applications have been investigated for more than decade with different results. ND particles are able to penetrate through the cell membrane and accumulate in the cytoplasm, where they form clusters surrounding cytoplasmic vesicles (I. Kratochvilova et al, J. Phys. Chem. C, 43 (2014) 25245-25252). The possibility that studied ND particles enter the cells should be discuss, also the final and real application of  LF-conjugated NDs  against oxidative stress, inflammatory response, and osteogenic differentiation of cells should be explained.    Also more effort should be given to details of biological and chemical aspects of LF-conjugated NDs  application.  

Author Response

Comments and Suggestions for Authors

Nanodiamond (ND) particles with various terminations for medical applications have been investigated for more than decade with different results. ND particles are able to penetrate through the cell membrane and accumulate in the cytoplasm, where they form clusters surrounding cytoplasmic vesicles (I. Kratochvilova et al, J. Phys. Chem. C, 43 (2014) 25245-25252). The possibility that studied ND particles enter the cells should be discuss, also the final and real application of LF-conjugated NDs against oxidative stress, inflammatory response, and osteogenic differentiation of cells should be explained. Also more effort should be given to details of biological and chemical aspects of LF-conjugated NDs application.  

Answer: Thank you for the reviewer’s comments. As reviewer suggested, the previous study demonstrated the intracellular uptake of NDs. Thus, we decide to cite the reference which is recommended by the reviewer. Also, we did not describe the possibility of the studied NDs in this study. As reviewer commented, we describe the possible application of the NDs in the last section of Discussion part.

Correction (Line 274~277): Previous study showed that ND particles can be internalized through the cell membrane and accumulate in the cytoplasm [31]. Consistent with the previous results, after 4 h incubation, FITC-conjugated NDs were observed around the cytoplasm and nuclei of cells (Fig. 4B).

Added reference:

[31] Kratochvilova, I.; Sebera, J.; Ashcheulov, P.; Golan, M.; Ledvina, M.; Micova, J.; Mravec, F.; Kovalenko, A.; Zverev, D.; Yavkin, B., et al. Magnetical and optical properties of nanodiamonds can be tuned by particles surface chemistry: Theoretical and experimental study. J Phys Chem C 2014, 118, 25245-25252.

Correction (Line 419~423): Consequently, this study demonstrated that NDs as a delivery carrier effectively ferry LF into cells. Through the effective delivery of LF with pleiotropic effects such as anti-inflammatory and anti-oxidant properties, LF-NDs could exert a synergistic effect on the osteogenic differentiation of MC3T3-El cells. Therefore, we expect that the anti-inflammatory and anti-oxidant LF-NDs will be applicable to treat bone tissue regeneration.

Reviewer 2 Report

This work investigates the production and characterizaion of lactoferrin conjugated nanodiamonds and their effects against oxidative stress, inflammatory response, proliferation and osteogenic differentiation of MC3T3-E1 cells (osteoblast-like cells).

In my opinion, the quality of the manuscript is high in terms of readibility, process explanations and characterizations performed, and average in terms of innovation.

Here some comments:

1) Resolution of Fig. 2 has to be increased.

General comment on NDs size distribution: have the authors performed the study with different NDs size? Does the size have an effect on functionalization and on cell internalization? Maybe this point should be faced and clarified. Are the authors using detonation NDs?

2) Table 1: oxygen concentration in XPS analysis is stable, in untreated and treated NDs. Is this reasonable? Surface composition has been modified, and probably did the amount of oxygen.

3) Fig. 7 can be optimized, reducing the space for legend and increasing the scale on the y-axys, in order to emphatyze the suppression of  IL-1β and TNF-α, since this achievement is one of the peculiarity of the manuscript.

4) Lines 299-300: why LPS increases the levels of pro-inflammatory cytokines? Is this an expected result?

(Absolutely yes, but a short description of the role of LPS would further increase the quality of the manuscript even for someone not in the field, as previously done all along the paper)

5) Lines 332-335: check the use of the terms osteoclasts and osteoblast.

6) Lines 414-415 must be removed, they're not part of the manuscript.

Author Response

Comment-1: Resolution of Fig. 2 has to be increased.

General comment on NDs size distribution: have the authors performed the study with different NDs size? Does the size have an effect on functionalization and on cell internalization? Maybe this point should be faced and clarified. Are the authors using detonation NDs?

Answer: Thank you for this comment. As reviewer indicated, we changed the Figure 2 with increased resolution. For the comment on NDs size effect, unfortunately, we could not perform the study with different NDs size and the effect on the functionalization and on cell internalization because we purchased the commercially available NDs product from Tokyo Chemistry Industry.   

Corrected Figure 2

Figure 2. (A) Particle size and distribution of NDs, LF (10 μg)-NDs, and LF (50 μg)-NDs measured by dynamic light scattering (DLS). (B) Attenuated total reflectance Fourier transform infrared (ATR-FTIR) spectra of NDs, LF (10 μg)-NDs, LF (50 μg)-NDs and Lactoferrin.

Comment-2: Table 1: oxygen concentration in XPS analysis is stable, in untreated and treated NDs. Is this reasonable? Surface composition has been modified, and probably did the amount of oxygen.

Answer: Thank you for this comment. As reviewer indicated, although the oxygen is similar among all groups, we think that LF is successfully immobilized on the surface of NDs because N contents are increased with increasing the loading amount of LF. Also, we agree that the XPS results are not enough to explain the surface immobilization of LF on the surface of NDs. Thus, we performed ATR-IF analysis and in vitro LF release from the NDs.

Correction (Line 251~254): NDs conjugated by LF (10 or 50 μg) were confirmed by increases in N1s component from 1.83% to 3.09% and 1.83% to 4.74, respectively, indicating that LF is existed on the surfaces of the NDs. To further confirm the LF immobilization on the NDs, ATR-FTIR spectra of each group are shown in Figure 2B before and after LF (10 or 50 μg) conjugation.  

Comment-3: Fig. 7 can be optimized, reducing the space for legend and increasing the scale on the y-axis, in order to emphasize the suppression of IL-1β and TNF-α, since this achievement is one of the peculiarity of the manuscript.

Answer: Thank you for the comment. As reviewer recommended, we corrected the reduction for legend as well as increase the scale on the y-axys in Figure 7.

Corrected Figure 7

Figure 7. (A) IL-1β and (B) TNF-α levels in cell supernatant secreted by LPS-stimulated cells treated with NDs with or without LF at 2, 6, 24, 72, and 120 h by ELISA.

Comment-4: Lines 299-300: why LPS increases the levels of pro-inflammatory cytokines? Is this an expected result?

(Absolutely yes, but a short description of the role of LPS would further increase the quality of the manuscript even for someone not in the field, as previously done all along the paper)

Answer: Yes, Lipopolysaccharide (LPS), major component of the outer membrane of Gram-negative bacteria, also known as lipoglycans and endotoxins. As previous studied, LPS-stimulated cells secreted increased amount of pro-inflammatory cytokines, such as TNF-α, IL-6, and IL-1β.

Added sentence (Line 396-400): In order to mimic the inflammatory environment in vitro, cells were treated with LPS which is a major component of the outer membrane of Gram-negative bacteria, also known as lipoglycans and endotoxins. As previously studied, LPS-stimulated cells secreted increased amount of pro-inflammatory cytokines, such as TNF-α, IL-6, and IL-1β [42, 43].

Added references:

[42] Kim, S.E.; Yun, Y.P.; Shim, K.S.; Jeon, D.I.; Park, K.; Kim, H.J. In vitro and in vivo anti-inflammatory and tendon-healing effects in achilles tendinopathy of long-term curcumin delivery using porous microspheres. J Ind Eng Chem 2018, 58, 123-130.

[43] Park, J.W.; Yun, Y.P.; Park, K.; Lee, J.Y.; Kim, H.J.; Kim, S.E.; Song, H.R. Ibuprofen-loaded porous microspheres suppressed the progression of monosodium iodoacetate-induced osteoarthritis in a rat model. Colloids Surf B Biointerfaces 2016, 147, 265-273.

Comment-5: Lines 332-335: check the use of the terms osteoclasts and osteoblast.

Answer: Thank you for the reviewer’s comment. We think that LF-NDs promoted the osteoblastic differentiation of MC3T3-El cells. We add this sentence in the text.

Addition (Line 338~339): These results supported that LF-NDs promoted the osteoblastic differentiation of MC3T3-El cells.

Comment-6: Lines 414-415 must be removed, they're not part of the manuscript.

Answer: Thank you for the comment. As reviewer indicated, we removed the line 414-415.

Round 2

Reviewer 1 Report

The manuscript was corrected in the required degree.